# Influence of Synthesis Conditions on the Crystal, Local Atomic, Electronic Structure, and Catalytic Properties of $(Pr_{1-x}Yb_x)_2Zr_2O_7$ ($0 \leq x \leq 1$) Powders

Victor V. Popov [1,2,*], Ekaterina B. Markova [3], Yan V. Zubavichus [4,*], Alexey P. Menushenkov [1], Alexey A. Yastrebtsev [1], Bulat R. Gaynanov [1], Olga V. Chernysheva [1], Andrei A. Ivanov [1], Sergey G. Rudakov [1], Maria M. Berdnikova [1], Alexander A. Pisarev [1], Elizaveta S. Kulikova [2], Nickolay A. Kolyshkin [2], Evgeny V. Khramov [2], Victor N. Khrustalev [3], Igor V. Shchetinin [5], Nadezhda A. Tsarenko [6], Natalia V. Ognevskaya [6] and Olga N. Seregina [6]

1. Department of Solid State Physics and Nanosystems, National Research Nuclear University MEPhI (Moscow Engineering Physics Institute), Moscow 115409, Russia; apmenushenkov@mephi.ru (A.P.M.); alexyastrebtsev@mail.ru (A.A.Y.); brgaynanov@gmail.com (B.R.G.); ovchernysheva@mephi.ru (O.V.C.); andrej.ivanov@gmail.com (A.A.I.); sgrudakov@mephi.ru (S.G.R.); mmberdnikova@mephi.ru (M.M.B.); aapisarev@mephi.ru (A.A.P.)
2. Kurchatov Synchrotron Radiation Source, National Research Center Kurchatov Institute, Moscow 123182, Russia; lizchkakul@mail.ru (E.S.K.); nickelprog@mail.ru (N.A.K.); evxramov@gmail.com (E.V.K.)
3. Department of Physical and Colloid Chemistry, Faculty of Science, RUDN University, Moscow 117198, Russia; ebmarkova@gmail.com (E.B.M.); vnkhrustalev@gmail.com (V.N.K.)
4. Synchrotron Radiation Facility SKIF, Boreskov Institute of Catalysis SB RAS, Koltsovo 630559, Russia
5. Material Science Department, National University of Science and Technology MISiS, Moscow 119049, Russia; ingvvar@gmail.com
6. JSC Design & Survey and Research & Development Institute of Industrial Technology, Moscow 115409, Russia; nadatsar@gmail.com (N.A.T.); ognevskayanv@mail.ru (N.V.O.); marioll1961@mail.ru (O.N.S.)
*   Correspondence: vvpopov@mephi.ru (V.V.P.); yvz@catalysis.ru (Y.V.Z.)

**Abstract:** The influence of $Yb^{3+}$ cations substitution for $Pr^{3+}$ on the structure and catalytic activity of $(Pr_{1-x}Yb_x)_2Zr_2O_7$ powders synthesized via coprecipitation followed by calcination is studied using a combination of long- (s-XRD), medium- (Raman, FT-IR, and SEM-EDS) and short-range (XAFS) sensitive methods, as well as adsorption and catalytic techniques. It is established that chemical composition and calcination temperature are the two major factors that govern the phase composition, crystallographic, and local-structure parameters of these polycrystalline materials. The crystallographic and local-structure parameters of $(Pr_{1-x}Yb_x)_2Zr_2O_7$ samples prepared at 1400 °C/3 h demonstrate a tight correlation with their catalytic activity towards propane cracking. The progressive replacement of $Pr^{3+}$ with $Yb^{3+}$ cations gives rise to an increase in the catalytic activity. A mechanism of the catalytic cracking of propane is proposed, which considers the geometrical match between the metal–oxygen (Pr–O, Yb–O, and Zr–O) bond lengths within the active sites and the size of adsorbed propane molecule to be the decisive factor governing the reaction route.

**Keywords:** praseodymium/ytterbium zirconates; crystal and local structures; catalytic cracking of propane; synchrotron XRD; X-ray absorption fine structure (XAFS); Raman spectroscopy; FT-IR spectroscopy

## 1. Introduction

In the last few decades, complex oxides $A_2B_2O_7$ (where A is typically a rare-earth cation in the oxidation state 3+, whereas B is a transition *d*-metal in the oxidation state 4+) with cubic pyrochlore, fluorite, or an intermediate structure derived from the former two have attracted vivid interest from researchers [1–6]. Primarily, they feature a rich

polymorphism representing a rather rare example of simultaneous disorder in both cation and anion sublattices upon a phase transition from fully ordered pyrochlore to a disordered defect fluorite structure [1–9]. Cubic pyrochlore complex oxides $A_2B_2O_7$ have also attracted much attention due to their ability to exhibit various types of geometrically frustrated magnetism [10,11]. Furthermore, complex oxide compounds and solid solutions from this family are prospective thermal barrier coatings [12–14], ion conductors [15,16], matrices for the immobilization of nuclear wastes [17,18], neutron-absorbing materials [19,20], and catalysts [21,22].

Nowadays, propane dehydrogenation is one of the main technologies for production of light olefins [23,24]. It has been shown that metal oxide-based materials can be used as promising catalysts for the conversion of propane to olefins [23–25]. More specifically, compounds with a common stoichiometry $(Pr_{1-x}Yb_x)_2Zr_2O_7$, synthesized at 1000 °C, have been recently successfully tested for the catalytic cracking of propane [26]. Unfortunately, this article does not provide details on the atomic structure of the polycrystalline materials used. Based on our previous studies [27,28], we consider all samples reported in [26] to have the same disordered defect–fluorite structure.

It is the cation radii ratio $\gamma = r_{A^{3+}}/r_{B^{4+}}$ that largely determines the crystal structure of $A_2B_2O_7$-type complex oxides. The $r_{A^{3+}}/r_{B^{4+}}$ radius–ratio threshold values are as follows: disordered fluorite $< 1.21 < \delta$-phase $< 1.42$–$1.44 <$ pyrochlore $< 1.78$–$1.83 <$ monoclinic pyrochlore $< 1.92$ [29]. It was shown that $Pr_2Zr_2O_7$ ($\gamma = 1.564$) has a pyrochlore structure (cubic, sp. gr. $Fd\bar{3}m$ (227)) [1,2,30]. In the case of preparing $Pr_2Zr_2O_7$ by the calcination of amorphous precursors, the pyrochlore phase is obtained through the formation of an intermediate fluorite phase (cubic, sp. gr. $Fm\bar{3}m$ (225)) [27]. The possibility of the formation of $Pr^{4+}$ cations in $Pr_{2\pm x}Zr_{2\pm x}O_{7\pm y}$ depending on the processing atmosphere and stoichiometry was shown in [31]. $Pr_2Zr_2O_7$ with a pyrochlore structure is a promising candidate for the quantum spin ices [10,32]. $Yb_2Zr_2O_7$ ($\gamma = 1.368$) can have both a defect fluorite structure [33] and a $\delta$-phase (rhombohedral, sp. gr. $R\bar{3}$ (148)) [28,30]. In the scientific literature on rare-earth zirconates, there are numerous reports on the use of chemical substitution either in the A [7,34,35] or B sites [36–38] or even in both the A and B sites simultaneously [39,40] to deliberately shift or control these phase transitions in other specific terms.

In this respect, it seems to be important to obtain deeper insights into the phase transitions occurring in praseodymium/ytterbium zirconates $(Pr_{1-x}Yb_x)_2Zr_2O_7$ ($0 \leq x \leq 1$) with exact stoichiometry as control parameter since functional crystalline starting and ending members of the series $Pr_2Zr_2O_7$ and $Yb_4Zr_3O_{12}$ are characterized by different syngonies. To the best of our knowledge, the $(Pr_{1-x}Yb_x)_2Zr_2O_7$ series has not been described in the literature so far. Herewith, the crystal, local atomic, and electronic structures of the synthesized samples were studied in detail using multiscale structural analysis, including a combination of diffraction, spectroscopy, and electron microscopy techniques [41,42]. In addition, the adsorption and catalysis-relevant properties of $(Pr_{1-x}Yb_x)_2Zr_2O_7$ polycrystalline materials synthesized at 1400 °C were elucidated, which helped us to establish mutual correlations between structural features of complex oxides $(Pr_{1-x}Yb_x)_2Zr_2O_7$ ($0 \leq x \leq 1$) and their catalytic characteristics.

## 2. Materials and Methods

### 2.1. Catalyst Synthesis

The complex oxides $(Pr_{1-x}Yb_x)_2Zr_2O_7$ ($0 \leq x \leq 1$) were prepared by a coprecipitation method [43]. The starting chemicals $Pr(NO_3)_3 \cdot 6H_2O$, $Yb(NO_3)_3 \cdot 4H_2O$ (with purity not lower than 99.95%), zirconium oxychloride octahydrate $ZrOCl_2 \cdot 8H_2O$ (99+%), and ammonium hydrate $NH_3 \cdot H_2O$ (analytical grade) were purchased from CHIMMED (Moscow, Russia) and used without additional purification. The starting salt solutions were prepared by mixing the initial reagents in the atomic ratios $[(1-x)Pr + xYb]:Zr = 1:1$ ($x = 0$, 0.25, 0.5, 0.75, and 1) followed by dissolution of the mixtures in distilled water. The mixed salt solutions were dropwise added into an $NH_3 \cdot H_2O$ aqueous solution under vigorous stirring. The as-

prepared suspensions (pH = 9.5–10.0) were aged for an hour at room temperature to ensure that the reaction is complete. The precipitates formed were filtered off, washed several times with distilled water, and then dried at 80 °C for 6 h. The dried precursors were then finely ground in an agate mortar and loaded into a muffle furnace LHT 02/16 (Nabertherm GmbH, Lilienthal, Germany). The powders were heated to a required temperature in the range of 600–1400 °C in air at a rate of 10 °C/min and then calcined isothermally for 3 h. The calcined samples were then cooled in a furnace to room temperature.

### 2.2. Characterization

The inductively coupled plasma atomic emission spectroscopy (ICP-AES) was used to quantify of the mass percentage of the metals in the synthesized precursors. ICP-AES measurements were carried out with a Vista-PRO spectrometer (Varian Inc., Palo Alto, CA, USA).

The operating conditions were as follows: auxiliary Ar flow rate 1.5 L/min; plasma Ar flow rate 13.5 L/min. All of the analyzed solutions were prepared by acid digestion. Samples ($\sim$100 mg, weighed with a precision of $\pm$0.1 mg) were dissolved in a mixture of analytical grade concentrated nitric acid (65% $m/m$; 3 mL), hydrochloric acid (37% $m/m$; 3 mL), and distilled water (15 mL) by boiling for 30 min. Cooled digested samples were further diluted with distilled water so that the concentration of the measured cations was 0.1–50 mg/L. The emission wavelengths were 422.293 and 410.072 nm for Pr; 218.572 and 222.447 nm for Yb; and 343.823 nm for Zr.

The simultaneous thermal analysis (STA) of the as-synthesized precursors involving thermogravimetry (TG) and differential scanning calorimetry (DSC) was carried out using a SDT Q600 analyzer (TA Instruments, New Castle, DE, USA) in a temperature range of 30–1400 °C at a heating rate of 10 °C/min in an air flow of 100 mL/min.

The X-ray diffraction (XRD) analysis of all of the synthesized samples was carried out on a laboratory MiniFlex 600 diffractometer (Rigaku, Tokyo, Japan) with monochromatized Cu $K_\alpha$-radiation ($\lambda$ = 1.5405 Å). All of the measurements were made at room temperature in the Bragg–Brentano geometry. The diffraction angle ($2\theta$) range was 10–100°, with a step size 0.025° and a dwell time of 3 s for each step. The operation X-ray source voltage and current were 40 kV and 30 mA, respectively. More detailed structural information was obtained from synchrotron X-ray powder diffraction (s-XRD) performed at the X-ray structural analysis beamline (Belok/XSA) of the Kurchatov Synchrotron Radiation Source (NRC Kurchatov Institute, Moscow, Russia) at the following parameters: the 2.5 GeV storage ring with an average current of 100 mA and the monochromatic radiation with a wavelength of 0.8 Å(the photon energy 15,498 eV) achieved using a Si(111) double-crystal monochromator. All of the measurements were made at room temperature in the Debye-Scherrer (transmission) geometry with an X-ray beam spot size of 400 μm [44]. The exposure time was 5 min. The tilt angle of the 2D Rayonix SX165 detector was 29.5°, with a sample-to-detector distance of 150 mm. The polycrystalline reference LaB$_6$ NIST SRM 660a sample was used for the calibration. Rietveld refinement of the XRD data was performed with Jana2006 software [45].

X-ray absorption spectra (XAFS) were measured at the BM25A-SpLine beamline [46] of the European Synchrotron Radiation Facility (ESRF) (Grenoble, France) and the Structural Materials Science beamline [47] of the Kurchatov Synchrotron Radiation Source (Moscow, Russia). EXAFS and XANES spectra were collected at the $K$-Zr (17,998 eV), $L_3$-Pr (5964 eV), and $L_3$-Yb (8944 eV) edges in the transmission mode at room temperature. The EXAFS spectra were analyzed using the VIPER [48] and IFEFFIT [49] program packages. The FEFF-9 [50] package was used to calculate the photoelectron backscattering amplitudes and phases. Initial structural models to be fit were constructed based on crystallographic results. The XANES spectra were fit using the XANDA program [48].

Raman spectra were collected using an inVia Qontor confocal Raman microscope (Renishaw plc, Wotton-Under-Edge, UK) ($\lambda$ = 532 nm) in a wavenumber range of 50–2700 cm$^{-1}$ with a spectral resolution of 1 cm$^{-1}$.

The Fourier transform infrared (FT-IR) spectra were recorded with a Nicolet iS50 FT-IR spectrometer (Thermo Fisher Scientific Inc., Waltham, MA, USA) ($\lambda$ = 1064 nm) in a wavenumber range of 400–4000 cm$^{-1}$ with a spectral resolution of 4 cm$^{-1}$.

Scanning electron microscopy (SEM) images were obtained on a Vega 3 scanning electron microscope (Tescan, Brno, Czech Republic). The electron beam energy was 30 keV, the working distance was 15 mm, and the spot size was approx. 50 nm. Energy dispersive X-ray spectroscopy (EDS) and elemental mapping images were obtained using an X-Act energy dispersive detector (Oxford Instruments, Oxford, UK) with a spectral resolution of 125 eV mounted on the SEM microscope. The characteristic X-ray radiation was automatically processed with the AZtec program.

The porosity parameters of the samples were determined from nitrogen vapor adsorption isotherms at 77 K measured with an ASAP 2020-MP (Micromeritics, Norcross, GA, USA) automatic high-vacuum setup in the relative pressure range from 0.001 to 0.98. The samples were preliminarily evacuated to a residual vacuum of below 7–10 Torr at 400 °C with a degassing time of 300 min.

A comparative quantification of the number of primary adsorption centers within the samples was attained. Water adsorption isotherms were measured at 293 K with a vacuum weighting unit with the McBain quartz spring balances (laboratory bench, Moscow, Russia) with a sensitivity of 10 µg for a weight of up to 100 mg.

The determination of acid sites was controlled by the UV absorption spectra of the catalytic systems exposed to pyridine. The concentrations were determined using a single-beam scanning spectrophotometer Agilent Cary 60 UV-Vis (Agilent Technologies, Santa Clara, CA, USA).

Catalytic tests for the cracking of propane were carried out over a temperature range of 100–900 °C with a step of 50 °C on a bench-top unit with a flow reactor. The catalysis was carried out at atmospheric pressure in a specially designed flow-through catalytic unit with a U-shaped quartz reactor under stationary conditions with a feed rate of 55.8 mmol/s. High purity propane (99.98 wt.%) was used as a feedstock. The reactor load was 0.05 g for all catalysts. The reaction was monitored at each temperature point using a Kristall 5000 M chromatograph equipped with a flame ionization detector and a thermal conductivity detector (Chromatek, Yoshkar-Ola, Russia).

## 3. Results and Discussion

### 3.1. Characterization of Catalytic Materials

The ICP-AES results showed that the experimental values of the chemical composition of the as-prepared precursors are close to the nominal stoichiometric ones (Table 1), which indicates that the intended complete incorporation of lanthanide and zirconium cations into the final solid samples has been successfully achieved.

**Table 1.** Chemical composition of synthetic precursors $(Pr_{1-x}Yb_x)_2Zr_2O_7$-prec as determined by alternative techniques.

| Sample $x$ | Stoichiometry | ICP-AES | EDS | $n$H$_2$O (TG at 200 °C) |
|---|---|---|---|---|
| 0 | Pr/Zr = 1/1 | Pr/Zr = 1/1.11 | Pr/Zr = 1/1.05 | 1.50 |
| 0.25 | Pr/Yb = 3/1 (Pr + Yb)/Zr = 1/1 | Pr/Yb = 2.78/1 (Pr + Yb)/Zr = 1/0.95 | | 1.76 |
| 0.5 | Pr/Yb = 1/1 (Pr + Yb)/Zr = 1/1 | Pr/Yb = 1.10/1 (Pr + Yb)/Zr = 1/0.95 | Pr/Yb = 0.98/1 (Pr + Yb)/Zr = 1/1.12 | 2.29 |
| 0.75 | Pr/Yb = 1/3 (Pr + Yb)/Zr = 1/1 | Pr/Yb = 1/3.11 (Pr + Yb)/Zr = 1/0.96 | | 2.47 |
| 1 | Yb/Zr = 1/1 | Yb/Zr = 1.06/1 | Yb/Zr = 1.16/1 | 1.85 |

SEM-EDS analysis was used to determine the particle morphology and draw element distribution maps related to the precursor particles. SEM images of the precursor samples

have shown that precursor particles are irregularly shaped aggregates with sizes spanning from a few to 10 μm and are composed of smaller primary particles (see Figure 1 and Figure S1 in Supplementary Materials). The Pr, Yb, Zr, and O element maps for $(Pr_{0.5}Yb_{0.5})_2Zr_2O_7$-prec samples demonstrate essentially homogeneous distribution for each element (Figure 1). These results suggest that Pr, Yb, Zr, and O atoms are evenly distributed and well dispersed on the surface of the as-prepared precursors. SEM-EDS tests were also performed with the purpose of obtaining a semiquantitative analysis of Pr, Yb, Zr, and O atomic concentrations. The calculated Pr/Yb and (Pr+Yb)/Zr atomic ratios are also listed in Table 1. These atomic ratios are close to the ICP-AES results and expected nominal stoichiometry. This solidly justifies the validity of the coprecipitation method for the reliable preparation of complex oxides with well controlled chemical composition.

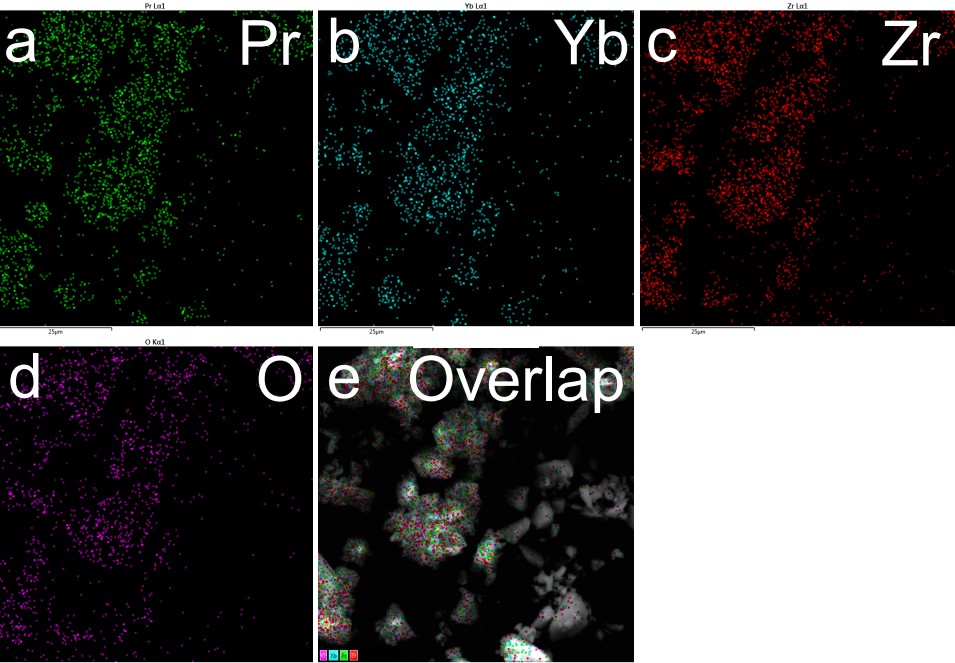

**Figure 1.** EDS map of Pr-$L_{\alpha 1}$ (**a**), Yb-$L_{\alpha 1}$ (**b**), Zr-$L_{\alpha 1}$ (**c**), O–$K_{\alpha 1}$ (**d**), and summary EDS/SEM image (**e**) for the $Pr_{0.5}Yb_{0.5}ZrOH$-prec sample.

All of the precursors were found to be X-ray amorphous irrespective of exact chemical composition. We used IR spectroscopy to probe the nature of functional groups present in the as synthesized precursors (Figure 2a).

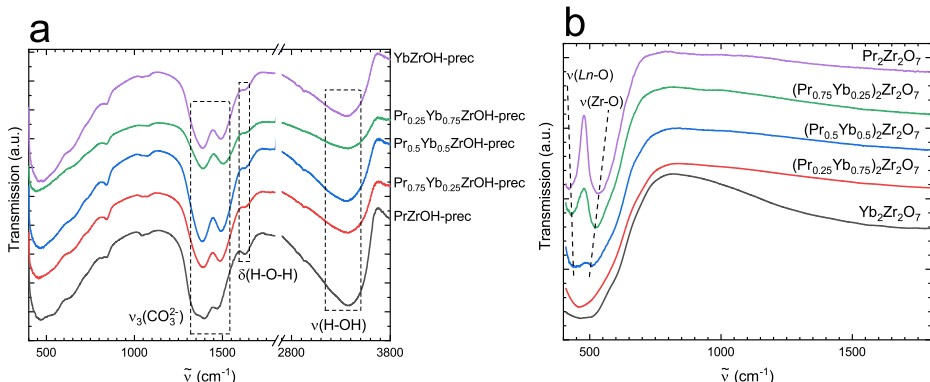

**Figure 2.** FT–IR spectra of the precursors (**a**) and the $(Pr_{1-x}Yb_x)_2Zr_2O_7$ powders synthesized at 1400 °C (**b**).

As can be seen from Figure 2a, the FT-IR spectra of precursors reveal several distinct absorption bands at about 465 (vs), 840 (s), 1045 (w), 1075 (w), 1395 (vs), 1472 (vs), 1626 (s), and 3370 cm$^{-1}$ (vs) (where vs—very strong, s—strong, and w—weak). Although it is difficult to assign each band to a specific compound, the very strong band at 465 cm$^{-1}$ may be associated with the O–*Ln*–O bending modes [14] or *Ln*–O vibration bands [51]. The band at about 840 cm$^{-1}$ is identified as the out-of-plane bending vibration of carbonate species. The band with components at about 1045 and 1075 cm$^{-1}$ corresponds to the $\nu_2$ symmetric stretching vibration of carbonate anions. The very strong doublet at 1395 and 1472 cm$^{-1}$ corresponds to the $\nu_3$ symmetric and asymmetric stretching vibrations, respectively. The observed peak-to-peak separation $\Delta\nu_3$ ($\nu_{as} - \nu_s$) is estimated to be about 80 cm$^{-1}$, which indicates the presence of metal-coordinated unidentate carbonate species [52]. The absorption band observed near 1626 cm$^{-1}$ and a broad strong band at ca. $\sim$3325 cm$^{-1}$ correspond to the bending mode ($\nu_2$) of H–O–H vibrations and longitudinal stretching vibrations of the O–H groups ($\nu_1$ and $\nu_3$) of the surface-adsorbed water molecules, respectively [51,53]. Moreover, a few weak bands at $\sim$1045–1075 cm$^{-1}$ may be also assigned to the bending vibrations of the M–O–H dringing hydroxyl group [53,54]. Therefore, the as-prepared precursors can be described as X-ray amorphous mixed lanthanide–zirconium hydrated hydroxycarbonates (Pr$_{1-x}$Yb$_x$)Zr(OH)$_{7-2y}$(CO$_3$)$_y \cdot n$H$_2$O. According to previous reports [54], the mean content of carbonate species in precursors of Gd titanates prepared by both sol-gel and coprecipitation methods was $\sim$12–15 wt.%. Based on the close chemical relation of these precursors and similar synthesis conditions, we formulate synthesized lanthanide–zirconium hydrated hydroxycarbonates as (Pr$_{1-x}$Yb$_x$)Zr(OH)$_5$(CO$_3$)$\cdot n$H$_2$O.

A self-consistent combined analysis of STA (Figure S2) and FT-IR (Figure S3) data enabled us to identify a few characteristic stages of transformations occurring with the precursors upon annealing. These include the removal of crystallization water ($n$H$_2$O) (up to 200–250 °C-see region I in Figure S2 and Table 1); the removal of hydroxyls; the partial decomposition of carbonate anions; crystallization (up to 600–800 °C-see regions II and III in Figure S2); the complete removal of residual carbonates; and the onset of structural phase transitions ($\geq$800 °C-see region IV in Figure S2). It is of note that the general sequence of these thermally activated transformation stages corresponds well to recently published experimental results on the synthesis of other types of complex oxides via coprecipitation [43,51,55,56].

Figure 3a depicts X-ray diffraction patterns of polycrystalline powders calcined at 1400 °C for 3 h. The phase composition and essential crystallographic parameters of the samples are compiled in Table 2.

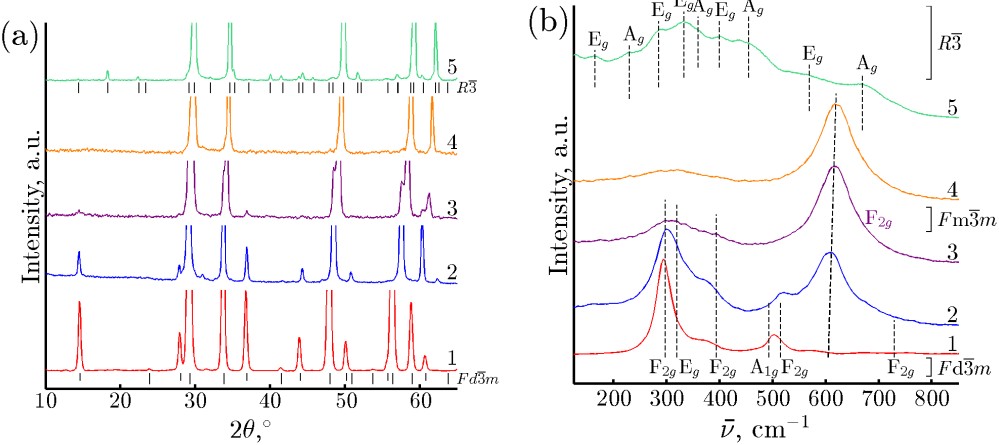

**Figure 3.** XRD patterns (**a**) and Raman spectra (**b**) of the (Pr$_{1-x}$Yb$_x$)$_2$Zr$_2$O$_7$ powders prepared by the calcination of precursors at 1400 °C/3 h: (1) $x = 0$; (2) $x = 0.25$; (3) $x = 0.5$; (4) $x = 0.75$; and (5) $x = 1$. The bars in (**a**) correspond to the reflexes of the pyrochlore (bottom) and $\delta$-phase (top) structures.

**Table 2.** The results of XRD Rietveld refinement for $(Pr_{1-x}Yb_x)_2Zr_2O_7$-1400 powders.

| Sample x | Structure (sp. gr.) | Phase % | Lattice Parameters, Å | Unit Cell Vol., Å³ | $x_{48f}$ | L, nm | $\epsilon$, % | $R_p$ | $R_{wp}$ % | $S_{GoF}$ |
|---|---|---|---|---|---|---|---|---|---|---|
| 0 | Cubic ($Fd\bar{3}m$) | 100 | $a = 10.7145(3)$ | 1230.03(5) | 0.3315 | 300(15) | 0.02(1) | 9.70 | 12.92 | 3.85 |
| 0.25 | Cubic ($Fd\bar{3}m$) | 100 | $a = 10.6338(3)$ | 1202.45(5) | 0.3381 | 245(12) | 0.28(3) | 4.89 | 6.54 | 2.21 |
| 0.5 | Cubic ($Fd\bar{3}m$) | 73 | $a = 10.6006(3)$ | 1191.22(5) | 0.3379 | >1000 | 1.24(5) | 6.70 | 8.49 | 1.15 |
| | Cubic ($Fm\bar{3}m$) | 27 | $a = 5.2431(3)$ | 114.13(5) | | 325(30) | 0.60(4) | | | |
| 0.75 | Cubic ($Fm\bar{3}m$) | 100 | $a = 5.2088(3)$ | 141.32(5) | | 235(12) | 0.12(3) | 7.52 | 10.18 | 1.47 |
| 1 | Rhomb. ($R\bar{3}$) | 100 | $a = 9.6641(9)$ $c = 8.9528(9)$ | 724.18(12) | | 104(5) | 0.27(7) | 6.03 | 8.92 | 5.30 |

$x_{48f}$—48f oxygen positional parameter (only for pyrochlore structure); $L$—the crystallite size; $\epsilon$—the microstrain value; and $S_{GoF}$—the goodness-of-fit.

As can be seen from Figure 3a and Tables 2 and S1, the type of the resultant crystal structure of $(Pr_{1-x}Yb_x)_2Zr_2O_7$ zirconates strongly depends on the composition as regards the lanthanide cations ($Ln^{3+} = Pr^{3+} + Yb^{3+}$). This ratio defines the effective rare-earth cation radius $r_{Ln^{3+}}$ and further the cation radii ratio $\gamma = r_{Ln^{3+}}/r_{Zr^{4+}}$. In the cases of $Pr_2Zr_2O_7$ ($\gamma = 1.564$) (JCPDS 20-1362, sp. gr. $Fd\bar{3}m$) and $(Pr_{0.75}Yb_{0.25})_2Zr_2O_7$ ($\gamma = 1.515$), the samples were characterized by the fcc pyrochlore-type structure. The intermediate sample $(Pr_{0.5}Yb_{0.5})_2Zr_2O_7$ ($\gamma = 1.467$) was a mixture of the pyrochlore and fluorite phases, whereas $(Pr_{0.25}Yb_{0.75})_2Zr_2O_7$ ($\gamma = 1.417$) remains the fluorite structure up to 1400 °C. In the case of $Yb_2Zr_2O_7$ ($\gamma = 1.368$), the rhombohedral $\delta$-phase (JCPDS 77-0739, sp. gr. $R\bar{3}$) is formed (Figure 3 and Tables 2 and S1). These results correspond well to reported literature data [27–29]. As is clearly shown in Tables 2 and S1, an increase in the content of $Yb^{3+}$ (0.985 Å) at the expenses of $Pr^{3+}$ (1.126 Å) gives rise to a decrease in the lattice parameters and unit cell volumes of the $(Pr_{1-x}Yb_x)_2Zr_2O_7$ complex oxides according to Vegard's law.

Fractional atomic coordinates, isotropic displacement parameters, and fractions of $Ln_{Zr} + Zr_{Ln}$ antisite pairs for the $(Pr_{1-x}Yb_x)_2Zr_2O_7$ samples calcined at 1400 °C are given in Table S2. For the samples $(Pr_{1-x}Yb_x)_2Zr_2O_7$ (x = 0.25 and 0.5) containing $Pr^{3+}$ and $Yb^{3+}$ cations simultaneously and dominated by the pyrochlore phase, it is $Zr^{4+}$ cations that substitute $Pr^{3+}$ ions in the *16d* site. Meanwhile, the $Zr^{4+}$ cations in the *16c* site are substituted only by $Yb^{3+}$ cations that have a smaller ion radius than $Pr^{3+}$. Note that the total concentration of oxygen vacancies in $(Pr_{1-x}Yb_x)_2Zr_2O_7$ is almost the same for both pyrochlore and fluorite structures. This parameter is equal to approximately one vacancy per seven filled oxygen positions. The change in the crystal structure changes only the spatial distribution of oxygen vacancies: from the precisely specified position of the oxygen vacancy *8a* (1/8, 1/8, 1/8) in pyrochlore (56 occupied oxygen positions out of 64 possible in the unit cell) to equally probable occupation of oxygen position *8c* (1/4, 1/4, 1/4) with coefficient 7/8 in fluorite.

Vibrational spectroscopy was used to study the oxygen-anion sublattices in the synthesized samples. Figure 3b shows Raman spectra of the $(Pr_{1-x}Yb_x)_2Zr_2O_7$ polycrystalline powders calcined at 1400 °C.

According to Figure 3b, the Raman spectrum of $Pr_2Zr_2O_7$-1400 reveals the following set of modes: 295 cm$^{-1}$ ($F_{2g}$), 310 cm$^{-1}$ ($E_g$), 370 cm$^{-1}$ ($F_{2g}$), 490 cm$^{-1}$ ($A_{1g}$), 500 cm$^{-1}$ ($F_{2g}$), and 755 cm$^{-1}$ ($F_{2g}$), evidencing the pyrochlore structure. These results correspond well to recently published literature data [28,57,58], which state that *Ln* zirconates with the pyrochlore structure (sp. gr $Fd\bar{3}m$) should demonstrate six active modes in Raman spectra. More specifically, five active modes ($A_{1g}$, $E_g$, and $3F_{2g}$) are due to vibrations of the oxygen O(1) atom located in the crystallographic site *48f*, and the remaining sixth active mode $F_{2g}$ is due to vibrations of the oxygen O(2) atom located in the crystallographic position *8b*. The

Raman spectrum of $(Pr_{0.75}Yb_{0.25})_2Zr_2O_7$-1400 is characterized by broadened pyrochlore phase-related modes as well as by an increased intensity of a mode peaked at 605 cm$^{-1}$ ($F_{2g}$ attributed to the fluorite-type structure). This indicates the onset of pyrochlore phase disordering, giving rise to the fluorite phase emergence. A further Yb$^{3+}$ cation substitution for Pr$^{3+}$ results in a complete disappearance of the disordered pyrochlore phase with the formation of somewhat defect fluorite-type structure in $(Pr_{0.25}Yb_{0.75})_2Zr_2O_7$-1400. The Raman spectrum of $Yb_2Zr_2O_7$-1400 is consistent with the dominant presence of the $\delta$-phase that has been descried by us earlier [28]. The Raman trends are in full accord with the aforementioned XRD conclusions (Table 2).

As it has been mentioned earlier (see Figure S3), an increase in the calcination temperature to 1400 °C strongly modifies the FT-IR spectra of $(Pr_{1-x}Yb_x)_2Zr_2O_7$ due to a variety of effects related to the occurrence of dehydration, the thermal decomposition of carbonate anions, crystallization, and phase transition processes. Figure 2b demonstrates FT-IR spectra of $(Pr_{1-x}Yb_x)_2Zr_2O_7$ powders prepared at 1400 °C.

In the FT-IR spectra of $(Pr_{1-x}Yb_x)_2Zr_2O_7$-1400 ($x = 0; 0.25$) samples with the pyrochlore structure (see Figure 2b and Table 2), the vibration band at 450–460 cm$^{-1}$ is distinctly split into two components: at 420–430 cm$^{-1}$ (*Ln*–O stretching vibration in the *Ln*–$O_8$ polyhedron [59] or the O–*Ln*–O bending mode [14,60]) and at 520–530 cm$^{-1}$ (Zr–O stretching vibrations in the Zr–$O_6$ unit [14,59,60]). For the similar spectrum of $(Pr_{0.5}Yb_{0.5})_2Zr_2O_7$-1400 encompassing both the pyrochlore (73%) and fluorite (27%) phases, this splitting is less pronounced. The respective components arise at 448 cm$^{-1}$ and 507 cm$^{-1}$. Meanwhile, the analogous vibration band shows up completely unsplit at 450–460 cm$^{-1}$ in the spectrum of $(Pr_{0.25}Yb_{0.75})_2Zr_2O_7$-1400, which is strictly single-phase defect fluorite (Figure 2b).

More detailed information on the local atomic structure around Zr$^{4+}$, Yb$^{3+}$, and Pr$^{3+}$ cations was retrieved using locally sensitive XANES (X-ray absorption near-edge structure) and EXAFS (extended X-ray absorption fine structure) X-ray absorption spectroscopy. XANES is especially suitable to probe changes in the electronic configuration of the central atom and symmetry of its nearest atomic surrounding.

Figure 4 shows XANES spectra measured at *K*-Zr, $L_3$-Yb, and $L_3$-Pr edges for the series of materials under study $(Pr_{1-x}Yb_x)_2Zr_2O_7$-1400. The Zr *K*-edge XANES spectra reveal two well-resolved near-edge peaks for all samples annealed at 1400 °C (see Figure 4a). However, at a closer inspection it becomes apparent that the specific values of the peak intensity ratio and peak splitting vary somewhat from one sample to another. The maximum peak intensity ratio ($I_B/I_A \sim 1.16$) and peak splitting ($\Delta E \sim 8.5$ eV) are observed for the two samples $(Pr_{1-x}Yb_x)_2Zr_2O_7$-1400 ($x = 0; 0.25$) that are dominated by the pyrochlore structure (see Figure 4b and Table 2). This strongly implies that the peak intensity ratio ($I_B/I_A$) and peak splitting ($\Delta E$) decrease gradually with an increase in the Yb$^{3+}$ content, i.e., upon a structural transition from pyrochlore ($x = 0; 0.25$) to defect fluorite ($x = 0.5; 0.75$) or $\delta$–phase ($x = 1.0$). Similar results were reported by other authors [61,62].

The exact position and shape of the main X-ray absorption peak, also referred to as the white line, in the Pr and Yb $L_3$-XANES spectra of $(Pr_{1-x}Yb_x)_2Zr_2O_7$-1400 samples, along with the small width of the white line (FWHM$\sim$6–7 eV), allow us to draw a conclusion regarding the occurrence of only trivalent *Ln*$^{3+}$ cations in these powders (Figure 4c,d) [63]. A closer inspection of the post-white line regions in the Pr and Yb $L_3$-XANES spectra reveals a characteristic X-ray absorption minimum with a split two-component structure, which can be regarded as a fingerprint of the pyrochlore structure [64]. This spectral feature is most pronounced in the Pr $L_3$-XANES spectra (see Figure 4c,d). The splitting becomes more smeared upon a structural phase transition "pyrochlore → fluorite" accompanying Yb$^{3+}$ substitution for Pr$^{3+}$.

The pseudo-radial distribution curves represented by Fourier transform (FT) moduli of EXAFS spectra measured at *K*-Zr, $L_3$-Yb, and $L_3$-Pr edges for the $(Pr_{1-x}Yb_x)_2Zr_2O_7$-1400 complex oxides are given in Figure 5. In the case of Zr *K*-edge spectra, all FT moduli of EXAFS spectra reveal an intense peak $\sim$1.7 Å (which is an interatomic distance uncorrected for the photoelectron backscattering phase shift) attributed to the first Zr–O coordination

sphere. At longer distances, somewhat weaker peaks are observed at ∼3.0–3.5 Å, corresponding to coordination shells encompassing metal cations. Real interatomic distances, coordination numbers, and Debye–Waller factors (DWF) extracted from EXAFS data quantitative analysis are listed in Table 3. One can see from Table 3 that the obtained interatomic distances are in good agreement with the results of XRD refinement.

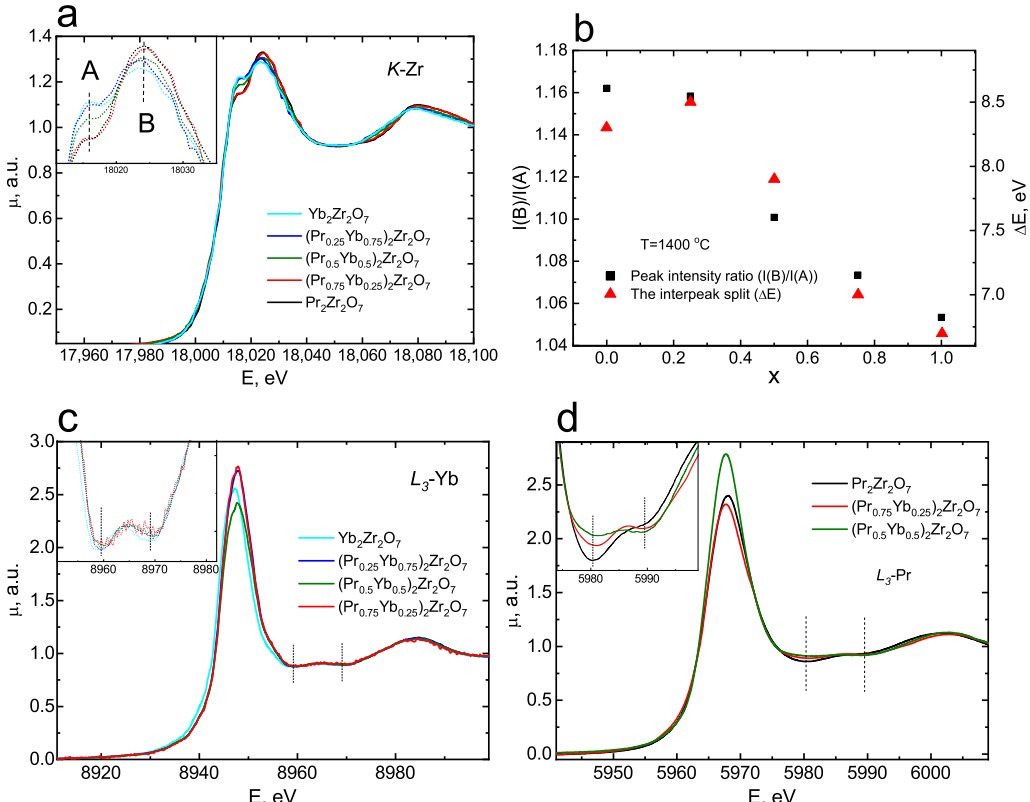

**Figure 4.** XANES spectra measured at *K*-Zr (**a**), $L_3$-Yb (**c**), and $L_3$-Pr (**d**) edges for the series of $(Pr_{1-x}Yb_x)_2Zr_2O_7$-1400 complex oxides; (**b**) the interpeak split ($\Delta E$) and peak intensity ratio ($I_B/I_A$) in Zr *K*-edge XANES spectra as a function of $Yb^{3+}$ content (*x*).

The long-distance FT peak of the Zr-*Ln* coordination shell grows weaker with an increase in the $Yb^{3+}$ content. This correlates with gradually increasing local disorder in the atomic environment of the $Zr^{4+}$ cations in the $(Pr_{1-x}Yb_x)_2Zr_2O_7$-1400 structure induced by the "pyrochlore → defect fluorite → $\delta$-phase" structural transitions.

One can see that the *Ln* $L_3$-edge EXAFS spectra are also sensitive to the appearance and further evolution of the pyrochlore phase enabled by an ongoing replacement of the lanthanide cations from Pr to Yb (Figure 5b,c). The FT moduli of Pr $L_3$-edge EXAFS spectra demonstrate an evident splitting of the first Pr–O coordination shell into two components corresponding to 2 shorter Pr–O(2) and 6 longer Pr–O(1) bonds for those $(Pr_{1-x}Yb_x)_2Zr_2O_7$ materials that possess the pyrochlore structure (*x* = 0, 0.25 and 0.5). The peak splitting in FT moduli of EXAFS spectra at Yb $L_3$-edge is weaker. Similar effects of the *Ln*–O coordination shell splitting were observed by us earlier for some *Ln* titanates [54,64], zirconates [64], and hafnates [43].

If we compare the local disordering trends observed in Pr and Yb $L_3$-edge XANES and EXAFS spectra (Figures 4 and 5) and the best-fit values of antisite defect concentrations calculated from powder diffraction (see Table S1), one can assume that the emerging fluorite phase should be enriched with Yb while Pr remains preferably in the pyrochlore phase. This means that the cation sublattice disordering upon the "pyrochlore → fluorite" transformation is initiated by a rearrangement involving the Yb sites.

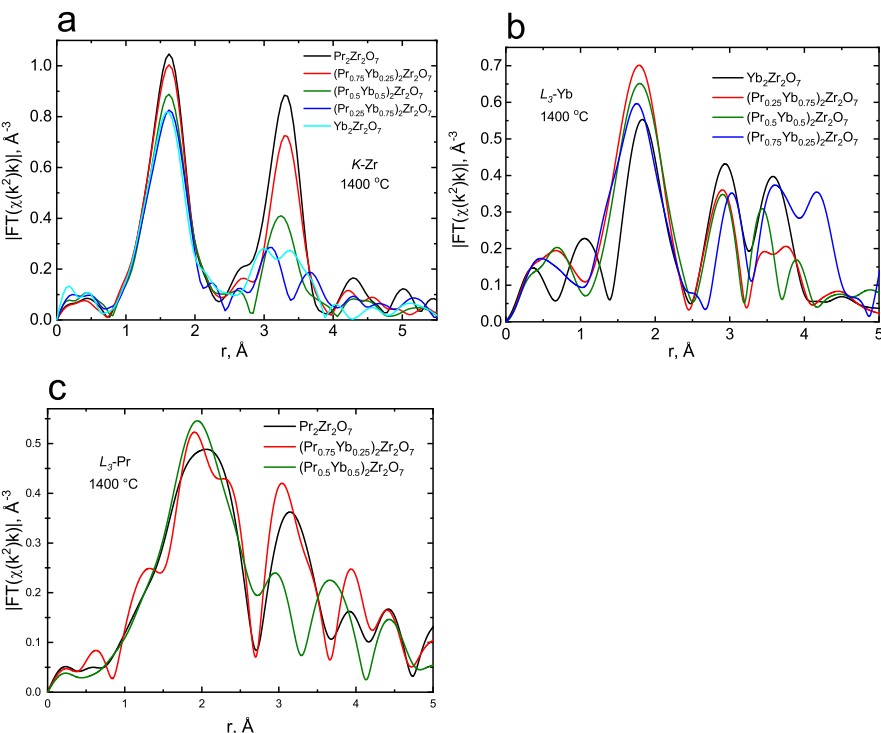

**Figure 5.** FT moduli of EXAFS spectra measured at *K*-Zr (**a**), $L_3$-Yb (**b**), and $L_3$-Pr (**c**) edges for the series of $(Pr_{1-x}Yb_x)_2Zr_2O_7$-1400 complex oxides.

**Table 3.** EXAFS fitting results for the Zr–O, Yb–O, and Pr–O shells of $(Pr_{1-x}Yb_x)_2Zr_2O_7$-1400 powders.

| *K*-Zr Edge | | | | | |
|---|---|---|---|---|---|
| **Sample** | **R(Zr–O), Å** | **$\sigma^2$(Zr–O), Å$^2$** | **N (sp. gr.)** | **$R_{dis}$, %** | **$R_{XRD}$, Å** |
| $Pr_2Zr_2O_7$ | 2.12(1) | 0.007(1) | 6 ($Fd\bar{3}m$) | 16 | 2.09(1) |
| $(Pr_{0.75}Yb_{0.25})_2Zr_2O_7$ | 2.11(1) | 0.006(1) | 6 ($Fd\bar{3}m$) | 15 | 2.10(1) |
| $(Pr_{0.5}Yb_{0.5})_2Zr_2O_7$ | 2.13(1) | 0.006(1) | 6.4 ($Fd\bar{3}m$ | 13 | 2.07(1) |
| | | | $+Fm\bar{3}m$) | | 2.27(1) |
| $(Pr_{0.25}Yb_{0.75})_2Zr_2O_7$ | 2.15(1) | 0.010(1) | 7 ($Fm\bar{3}m$) | 17 | 2.26(1) |
| $Yb_2Zr_2O_7$ | 2.24(1) | 0.008(1) | 6 ($R\bar{3}$) | 15 | 2.13(1) |

| $L_3$-Yb edge | | | | | |
|---|---|---|---|---|---|
| **Sample** | **R(Yb–O), Å** | **$\sigma^2$(Yb–O), Å$^2$** | **N (sp. gr.)** | **$R_{dis}$, %** | **$R_{XRD}$, Å** |
| $(Pr_{0.75}Yb_{0.25})_2Zr_2O_7$ | 2.14(1) | 0.008(1) | 2 ($Fd\bar{3}m$) | 9 | 2.05(1) |
| | 2.31(1) | 0.004(1) | 6 | | 2.31(1) |
| $(Pr_{0.5}Yb_{0.5})_2Zr_2O_7$ | 2.23(1) | 0.012(1) | 6.2 ($Fd\bar{3}m$ | 12 | 2.07(1) |
| | | | $+Fm\bar{3}m$) | | 2.27(1) |
| $(Pr_{0.25}Yb_{0.75})_2Zr_2O_7$ | 2.25(1) | 0.010(1) | 7 ($Fm\bar{3}m$) | 16 | 2.26(1) |
| $Yb_2Zr_2O_7$ | 2.23(1) | 0.013(1) | 6 ($R\bar{3}$) | 9 | 2.28(2) |

| $L_3$-Pr edge | | | | | |
|---|---|---|---|---|---|
| **Sample** | **R(Pr–O), Å** | **$\sigma^2$(Pr–O), Å$^2$** | **N (sp. gr.)** | **$R_{dis}$, %** | **$R_{XRD}$, Å** |
| $Pr_2Zr_2O_7$ | 2.28(2) | 0.015(2) | 2 ($Fd\bar{3}m$) | 15 | 2.32(1) |
| | 2.53(2) | 0.022(2) | 6 | | 2.61(1) |
| $(Pr_{0.75}Yb_{0.25})_2Zr_2O_7$ | 2.16(2) | 0.015(2) | 2 ($Fd\bar{3}m$) | 16 | 2.30(1) |
| | 2.44(2) | 0.019(2) | 6 | | 2.55(1) |
| $(Pr_{0.5}Yb_{0.5})_2Zr_2O_7$ | 2.39(2) | 0.011(2) | 2 ($Fd\bar{3}m$) | 18 | 2.29(1) |
| | 2.56(2) | 0.020(2) | 6 | | 2.58(1) |
| $(Pr_{0.25}Yb_{0.75})_2Zr_2O_7$ * | | | | | 2.26(1) |

Values given in parentheses correspond to the estimated standard deviations (esd); N is the fixed coordination number; R is the interatomic distance from EXAFS; $\sigma^2$ is the Debye–Waller factor (DWF); $R_{dis}$ is the discrepancy index; $R_{XRD}$ is the interatomic distance from XRD; and *—only XRD data available.

### 3.2. Catalytic Properties

According to the literature data [65,66], M (metal)-O (oxygen) bond lengths, the presence of oxygen vacancies, and the active metal type are the three main factors determining catalytic properties of complex oxide compounds. As it was mentioned before in Section 3.1, the progressive $Yb^{3+}$ substitution for $Pr^{3+}$ in $(Pr_{1-x}Yb_x)_2Zr_2O_7$ ($0 \leq x \leq 1$) polycrystalline materials induces prominent changes both in crystallographic and local-structure parameters. This means that all three aforementioned factors are involved. Therefore, it might be anticipated that all $(Pr_{1-x}Yb_x)_2Zr_2O_7$ polycrystalline materials would manifest different catalytic activities.

Based on earlier studies of catalytic propane dehydration, active sites in $(Pr_{1-x}Yb_x)_2Zr_2O_7$ were envisaged as a couple of coordinatively unsaturated cations ($Pr^{3+}/Yb^{3+}$ and $Zr^{4+}$) bound to an adjacent oxygen vacancy and lattice oxygen. Unfortunately, such a suggestion was not supported by structural studies of the catalytic systems, prepared at 1000 °C [26]. This temperature is not high enough for phase transitions to take place in *Ln* zirconates [27,28], and the studied samples obviously had a disordered defect–fluorite structure.

The canonical mechanism of alkane cracking implies the detachment of a hydrogen atom from the alkane-derived carbocation, giving rise to another carbocation, which is further decomposed via the $\beta$-scission (i.e., the scission of the C–C bond in the $\beta$-position with respect to the carbon atom bearing the positive charge in the carbocation). In this process, carbocations act as chain carriers. Typically, carbocations formed from components of the crude feedstock are prominently stable. In particular, carbocations can be formed upon the catalyst-assisted protonation of alkenes. Alkenes can be present in the feedstock as admixtures. Alternatively, alkenes can be generated from alkanes via the free radical cracking (provided that the reaction temperature is sufficiently high) or via the hydrogen detachment by Lewis acid sites of the catalyst (dehydrogenation) [23,67].

Herewith, we elucidate the catalytic properties of $(Pr_{1-x}Yb_x)_2Zr_2O_7$ polycrystalline materials prepared by the calcination at 1400 °C.

According to previous reports, the thermally activated propane cracking starts above 500 °C. The apparent propane conversion is as low as 2% at 600 °C, but it increases to 20% at 700 °C. The major reaction products were identified as methane and ethane [68]. In the presence of $(Pr_{1-x}Yb_x)_2Zr_2O_7$-1400 catalysts, the propane conversion increases from 65% to 94% at 700 °C on going from $Pr_2Zr_2O_7$ to $Yb_2Zr_2O_7$ (Figure 6).

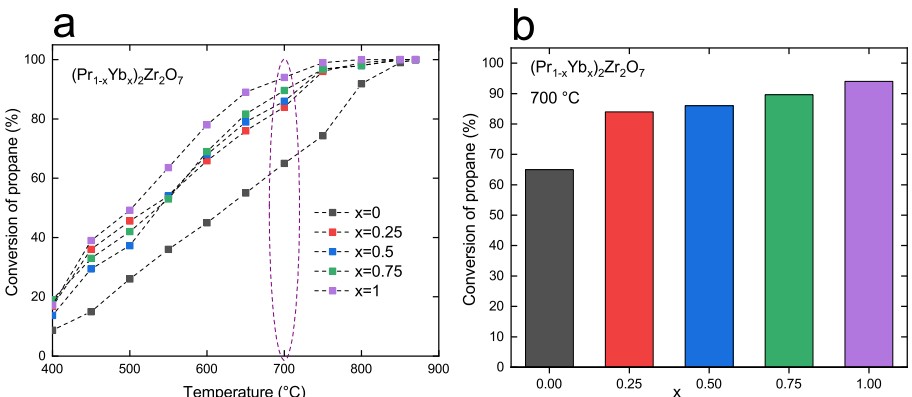

**Figure 6.** Temperature dependence of conversion of propane for $(Pr_{1-x}Yb_x)_2Zr_2O_7$-1400 catalysts with different values of $x$ (**a**); propane conversion at the optimum cracking temperature (**b**).

As it can be seen in Figure 6a, the propane conversion reaches the values of 100% at 850 °C for all catalytic systems under study. If we compare these results with earlier studies of thermally activated cracking [68], the same level of conversion is achieved at systematically lower temperatures. Furthermore, the selectivity to ethylene and propylene is also altered (Figure 7).

Opposite trends in ethylene and propylene selectivity are observed on going from $Pr_2Zr_2O_7$ to $Yb_2Zr_2O_7$ (Figure 7a–d). The decrease in ethylene selectivity along the series $Pr_2Zr_2O_7 > (Pr_{0.75}Yb_{0.25})_2Zr_2O_7 > (Pr_{0.5}Yb_{0.5})_2Zr_2O_7 > (Pr_{0.25}Yb_{0.75})_2Zr_2O_7 > Yb_2Zr_2O_7$ strongly implies that the pyrochlore-type structure primarily catalyzes the propane decomposition to methane and ethylene (Figure 7a,b). Meanwhile, the increase in the propylene yield along the series $Pr_2Zr_2O_7 < (Pr_{0.75}Yb_{0.25})_2Zr_2O_7 < (Pr_{0.5}Yb_{0.5})_2Zr_2O_7 < (Pr_{0.25}Yb_{0.75})_2Zr_2O_7 < Yb_2Zr_2O_7$ suggests that propane dehydrogenation predominantly proceeds over defect fluorite and $\delta$-phase (Figure 7c,d). Therefore, distinctly different types of catalytically active sites towards the propane cracking emerge in the complex oxide systems under study, which are characterized by specific features of the crystal and local atomic structures.

To gain more insight into the nature of catalytic activity of the systems under study, the electron acceptor properties of the catalysts' surface were evaluated. The mean number (N) and strength ($E_0$) of Lewis acid sites were calculated based on kinetics pyridine accumulation (Table 4). The bimodal character of pyridine adsorption curves is of note. This evidently means that there are several distinct adsorption sites. We tentatively assign them to Zr–O, Pr–O, and Yb–O active sites. The number of electron acceptor sites increases along the series $Pr_2Zr_2O_7 < (Pr_{0.75}Yb_{0.25})_2Zr_2O_7 < (Pr_{0.5}Yb_{0.5})_2Zr_2O_7 < (Pr_{0.25}Yb_{0.75})_2Zr_2O_7 < Yb_2Zr_2O_7$ (see Table 4).

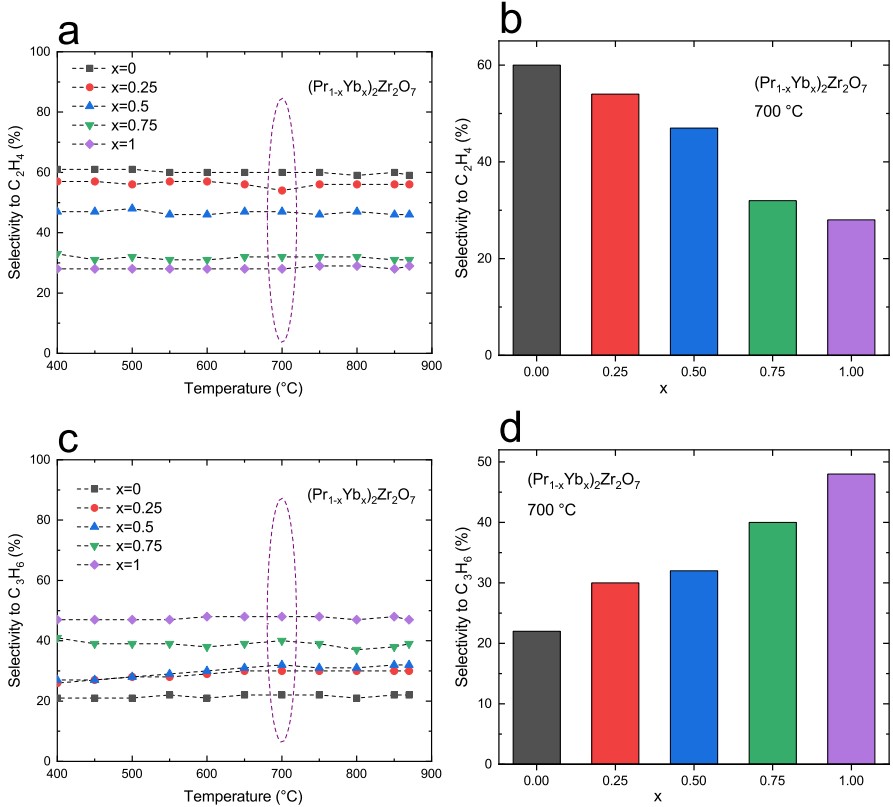

**Figure 7.** Selectivity to ethylene (**a**,**b**) and to propylene (**c**,**d**) for complex oxide catalysts $(Pr_{1-x}Yb_x)_2Zr_2O_7$-1400 with different values of *x*; (**b**,**d**)-selectivity at the optimum cracking temperature.

**Table 4.** Surface morphology characteristics of $(Pr_{1-x}Yb_x)_2Zr_2O_7$-1400 catalysts.

| Catalyst | PAC, mmol/g | N, μmol/g | $W_{C_3H_8}$ cm³/g | $E_0$, kJ/mol | $S_{BET}$, m²/g | $r$, nm |
|---|---|---|---|---|---|---|
| $Pr_2Zr_2O_7$ | 63.3 | 288 | 0.004 | 27.2 | 11 | 11.7 |
| $(Pr_{0.75}Yb_{0.25})_2Zr_2O_7$ | 55.8 | 298 | 0.007 | 31.1 | 7 | 10.0 |
| $(Pr_{0.5}Yb_{0.5})_2Zr_2O_7$ | 48.5 | 304 | 0.008 | 33.1 | 6 | 9.6 |
| $(Pr_{0.25}Yb_{0.75})_2Zr_2O_7$ | 44.1 | 309 | 0.008 | 35.7 | 5 | 9.0 |
| $Yb_2Zr_2O_7$ | 41.3 | 311 | 0.009 | 37.7 | 3 | 8.3 |

PAC is the number of primary adsorption centers calculated from comparative isotherm of water vapor adsorption; N is the total number of electron acceptor centers (Lewis acid sites) calculated from pyridine adsorption curves; $W_{C_3H_8}$ is the volume of adsorbed propane calculated according to the Barrett, Joyner, and Halenda (BJH) method; $E_0$ is the characteristic adsorption energy calculated according to the Dubinin-Astakhov method; $S_{BET}$ is the BET specific surface area; and $r$ is the mean pore size in the capillary condensation mode determined by the BJH method.

Primary adsorption centers (or PAC) were quantified in order to establish the dominant mechanism of propane catalytic conversion. The number of primary adsorption centers and total number of oxygen-containing centers were determined using the adsorption of water vapor from the gaseous phase (Table 4). The latter value increases along the series $Pr_2Zr_2O_7 < (Pr_{0.75}Yb_{0.25})_2Zr_2O_7 < (Pr_{0.5}Yb_{0.5})_2Zr_2O_7 < (Pr_{0.25}Yb_{0.75})_2Zr_2O_7 < Yb_2Zr_2O_7$.

The textural properties of the $(Pr_{1-x}Yb_x)_2Zr_2O_7$-1400 catalysts were analyzed using the $N_2$ adsorption–desorption isotherms with the explicit account for the pore size distribution. All of the materials under study were characterized by a non-porous surface with a moderately low concentration of mesopores. The mean pore size decreases along the series $Pr_2Zr_2O_7 < (Pr_{0.75}Yb_{0.25})_2Zr_2O_7 < (Pr_{0.5}Yb_{0.5})_2Zr_2O_7 < (Pr_{0.25}Yb_{0.75})_2Zr_2O_7 < Yb_2Zr_2O_7$ (Table 4).

The $(Pr_{1-x}Yb_x)_2Zr_2O_7$-1400 catalysts exhibiting high activity and selectivity towards propane cracking possess active sites constructed of pairs of zirconium and lanthanoid cations bound to oxygen vacancies and lattice oxide anions. These sites are capable of efficient C–H bond activation, which is actually the rate-limiting step in light alkane dehydrogenation [25].

Typically, the specific surface area strongly affects the activity of heterogeneous catalysts. In our case, the total propane conversion demonstrates a prominent tend to increase with a decrease in the specific surface area of the catalysts from 11 m²/g down to 3 m²/g on going from $Pr_2Zr_2O_7$ to $Yb_2Zr_2O_7$ (Figure 8a). This clearly implies that the available surface exerts the minimum influence on the apparent activity of the catalysts towards the propane dehydrogenation. Such behavior is very unusual and thus should be elucidated in more detail. We assert that the total and specific activities pass through the maxima with a very small degree of filling. Each real surface of these catalytic systems is characterized by a blocky, mosaic structure, as a result of which isolated migration regions may appear on the surface, separated from each other by energy or geometric barriers. They can be similar to real cells, for example, faces of elementary crystals adjacent to adsorption centers with an increased adsorption potential characteristic of an energetically inhomogeneous surface. Cracks and other surface disturbances, crystal defects, and stoichiometric composition disorders can also cause migration areas. This leads to the appearance of random catalytic centers. If there were a large number of them, we would not be able to unambiguously characterize the mechanism of the processes taking place. For the formation of active centers in the form of a cluster of n-atoms, it does not matter what the origin and nature of these inhomogeneities are. It is important that when forming a layer of catalytic centers, the surface (which is very insignificant) allows for the free migration of particles only in limited areas. These surface disturbances are an obstacle to free movement through the volume of the catalyst, forming potential pits where additional catalytic centers should accumulate. In our case, their number is very small. The catalyst surface is a collection of closed migration regions. Experimental data on the sintering of catalysts were a confirmation of these ideas. The rate of deactivation of catalysts obeys the first-order equation for

the concentration of additional centers on the surface. Consequently, the additional centers on the surface do not depend on each other and do not interact with each other. The size of the migration areas significantly exceeds the radius of action of molecular forces (by tens of times). Consequently, the formation of additional centers of catalytic cents is an independent event. The distribution of these cents over the surface of the catalyst obeys the law of chance and tends to a minimum. Thus, we can make an unambiguous conclusion that the main catalytic centers are precisely the Lewis acid centers.

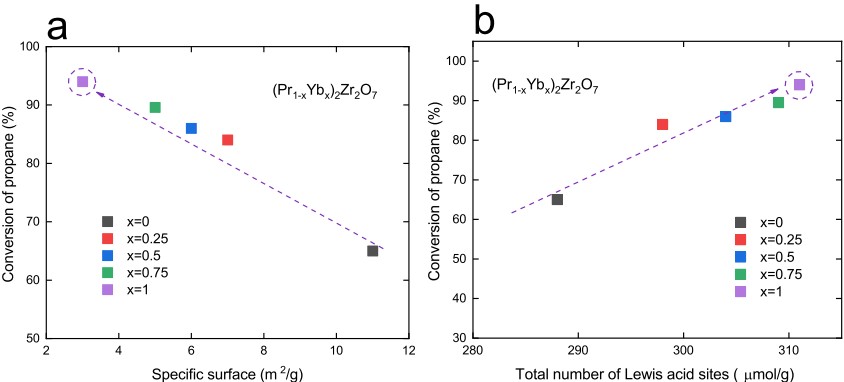

**Figure 8.** Apparent conversion of propane as a function of (**a**) specific surface area of the catalysts and (**b**) total number of Lewis acid sites.

Meanwhile, an increase in the number of Lewis acid sites with increasing Yb content in $(Pr_{1-x}Yb_x)_2Zr_2O_7$ gives rise to a virtually linear increase in the propane conversion. Therefore, it is the number of Lewis acid sites that plays a decisive role in determining the activity of the catalytic systems under study (Figure 8b).

According to Jeon et al. [69], for the non-oxidative dehydrogenation of propane on metal oxides the propane molecules are adsorbed on $M^{x+}$ surface sites. Then, hydrogen from the adsorbed propane molecule is abstracted by surface oxygen, forming a hydroxyl group and a propyl species.

Proceeding to the analysis of product ratio, we should note that the $Zr^{4+}$ transition metal cations are capable of propane activation due to the completely filled $3d^{10}$ orbitals and unoccupied 4th electronic shell. The metal ion's electron affinity becomes maximized when the 4th electronic shell is completely empty. In addition to $Zr^{4+}$, the complex oxides $(Pr_{1-x}Yb_x)_2Zr_2O_7$ contain rare-earth cations with different configurations of $4f$ electrons. The empty shells are prone to electron capture from adsorbates, which can serve as a mechanism of adsorption activation over this metal ion active site. It is important that the dominant phase switches from pyrochlore through defect fluorite to $\delta$-phase on going from $Pr_2Zr_2O_7$ to $Yb_2Zr_2O_7$, which is accompanied by specific changes in the crystal and local atomic structures of the materials (*vide supra*, Section 3.1). In particular, pyrochlore phases are abundant with oxygen vacancies. Probably, these vacancies can also act as acceptors of the electron density of adsorbate molecules.

The established correlation between catalytic behavior and structural features of the $(Pr_{1-x}Yb_x)_2Zr_2O_7$ complex oxides can be rationalized in the following way.

Carbon atoms in the propane molecule bear an excessive electron density due to the induction effect. The $C_3H_8$ molecules become adsorbed at the gas–solid interface on the Lewis-cationic ($Pr^{3+}/Yb^{3+}$) and $Zr^{4+}$ sites of the catalysts. An increase in the fraction of $Yb^{3+}$ cations (characterized by a higher ion potential ($Z/r$) with respect to $Pr^{3+}$ in $(Pr_{1-x}Yb_x)_2Zr_2O_7$ increases both the number (N) and strength ($E_0$) of Lewis acid surface sites, which promotes propane adsorption ($W_{C_3H_8}$) (see Table 4). This ultimately results in enhanced propane conversion, i.e., catalytic activity (see Figures 6 and 8b).

The selectivity to $C_2H_4$ decreases from 60% to 28% on shifting from $Pr_2Zr_2O_7$ to $Yb_2Zr_2O_7$ (see Figure 7a,b), i.e., the tendency of propane decomposition via the C–C bond

scission is weakened. A tentative mechanism of propane decomposition yielding ethylene is shown in Figure 9. Such a scheme is well supported by the structural parameters of $Pr_2Zr_2O_7$. The size factor (metal–oxygen bond length R(Pr–O(1)) = 2.53(2) Å), the abundance of oxygen vacancies (8*a* sites in the pyrochlore structure), and the availability of vacant $4f$ orbitals in $Pr^{3+}$ ($4f^2$) together promote the horizontal orientation of adsorbed propane molecules, in which the distance between terminal carbon atoms is 2.51 Å. The decomposition of propane occurs via the C–C bond scission over $Pr_2Zr_2O_7$. Regarding $Yb_2Zr_2O_7$, the metal–oxygen bonds (Pr/Yb)–O become shorter, which gives rise to a switch of the dominant propane cracking mechanism to the C-H on scission, i.e., dehydrogenation (Figure 10).

According to Figure 7c,d discussed earlier, the selectivity to $C_3H_6$ increases from 20% to 47% on going from $Pr_2Zr_2O_7$ to $Yb_2Zr_2O_7$. This means that the C–H bond scission yielding propylene tends to dominate. A tentative mechanism of the dehydrogenation in that case is shown in Figure 10. The propane dehydrogenation predominance is probably due to a specific orientation of the adsorbed propane molecule on active sites of the $(Pr_{1-x}Yb_x)_2Zr_2O_7$ complex oxide catalysts. For instance, the metal–oxygen bond lengths are R(Zr–O) = 2.15(1) Å, R(Pr–O) = 2.26(1) Å and R(Yb–O) = 2.25(1) Å in the $(Pr_{0.25}Yb_{0.75})_2Zr_2O_7$-1400 sample characterized by the defect fluorite structure. Moreover, analogous values are R(Zr–O) = 2.24(1) Å and R(Yb–O) = 2.23(1) Å for $Yb_2Zr_2O_7$, which features the $\delta$-phase (see Table 3). In both the above cases, the size factor prevents the horizontal adsorption of the propane molecule. Instead, the propane molecule is obliged to adsorb vertically. Such an adsorbate orientation on Zr–O and (Pr/Yb)–O active sites promotes efficient C–H bond scission in $C_3H_8$ to afford M–$C_3H_7$ and O–H surface complexes. Ultimately, gaseous $C_3H_6$ and $H_2$ are released as a result of the $\beta$-hydrogen elimination from M–$C_3H_7$ (Figure 10). It should be noted that the proposed propane dehydrogenation mechanism is in good agreement with that described in the literature [69].

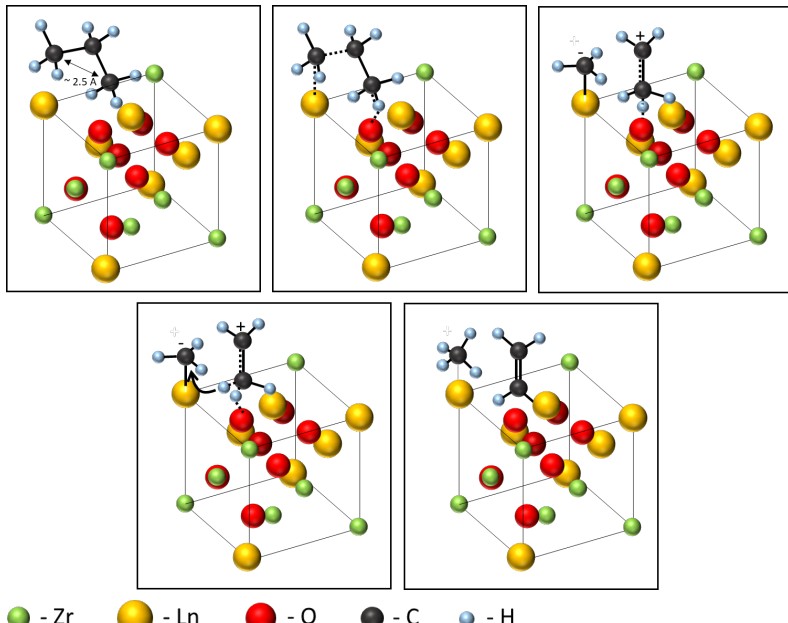

**Figure 9.** The mechanism of propane decomposition yielding ethylene.

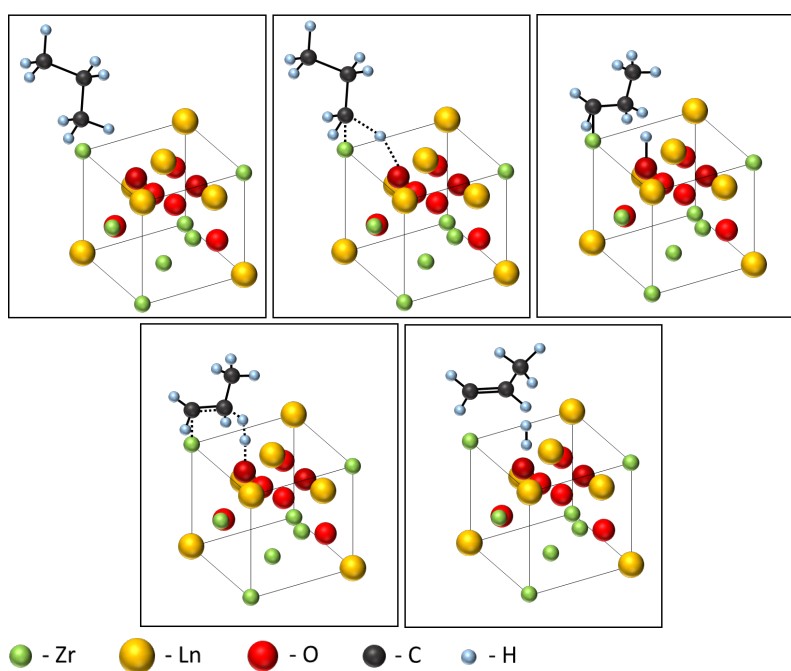

**Figure 10.** The mechanism of propane decomposition yielding propylene.

Therefore, all of the surface atoms of the catalyst could similarly promote dehydrogenation. Meanwhile, only specific configurations with enlarged interatomic separations could activate side reactions (e.g., decomposition through the C–C bond scission). Thus, the geometrical and electronic factors of the catalyst surface structure govern its activity. This could be rationalized recoursing to the concept of non-specific (structure-insensitive) and specific (structure-sensitive) reactions. Dehydrogeation reactions, being typical structure-insensitive ones, could be just as efficiently catalyzed by active sites with either short or very long interatomic distances. On the contrary, propane decomposition via the C–C bond scission is a typical structure-sensitive reaction that could proceed only on appropriately organized active sites with specific geometry featuring interatomic distances of ca. 2.5 Å.

Apparent activation energies of the reaction were calculated for all the complex oxide catalysts under study (Table 5), postulating that the propane cracking is the first-order reaction [68]. The catalyst-free thermally activated propane cracking is characterized by the activation energy of 104 kJ/mol [68]. The activation energy is decreased in the presence of catalysts by a few tens kJ/mol, which facilitates the process a lot.

**Table 5.** Major catalysis-relevant characteristics of $(Pr_{1-x}Yb_x)_2Zr_2O_7$-1400 materials.

| Catalyst | TON $\times 10^6$ | $E_a$, kJ/mol/g | CB, % |
|---|---|---|---|
| $Pr_2Zr_2O_7$ | 4 | 89 | 97 |
| $(Pr_{0.75}Yb_{0.25})_2Zr_2O_7$ | 2.7 | 87 | 97 |
| $(Pr_{0.5}Yb_{0.5})_2Zr_2O_7$ | 2.5 | 86 | 98 |
| $(Pr_{0.25}Yb_{0.75})_2Zr_2O_7$ | 2.1 | 79 | 98 |
| $Yb_2Zr_2O_7$ | 2 | 77 | 97 |

TON-turnover number; $E_a$—propane cracking activation energy; and CB—carbon balance.

Importantly, the activation energy for the catalysts under study remain essentially constant over the entire temperature range, which means that the reaction obeys the same carbenium mechanism and proceeds in the catalyst-assisted heterogeneous mode rather than switching to the homogeneous gas-phase mode.

The propane cracking could yield not only $C_xH_y$ radicals but free carbon as well, which could in its turn react with hydrogen, affording methane and other hydrocarbons.

With an increase in temperature, the reaction between free carbon atoms starts to prevail, which gives rise to the coking of available catalytic sites (Figure 11).

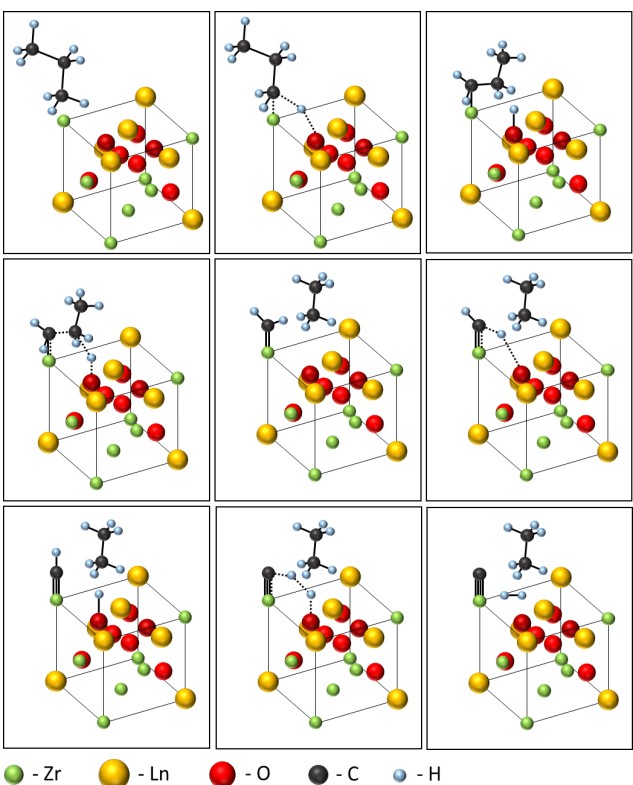

● - Zr　　● - Ln　　● - O　　● - C　　● - H

**Figure 11.** The mechanism of coking of $(Pr_{1-x}Yb_x)_2Zr_2O_7$-1400 catalysts.

The experimentally observed decrease in the activity of the cracking catalysts can be due to the shielding of the active surface with soot deposits (Figure 12). According to Figure 12, the most prominent loss of activity occurs after the 5th cycle for all $(Pr_{1-x}Yb_x)_2Zr_2O_7$ catalysts under study.

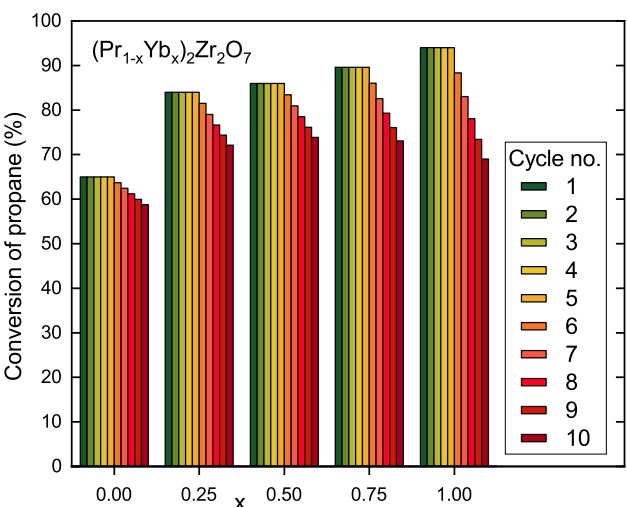

**Figure 12.** Catalytic performance degradation of the $(Pr_{1-x}Yb_x)_2Zr_2O_7$-1400 systems under cycling.

As it can be judged from Figure 11, the propane molecule should be adsorbed vertically on an active site in order to enable the coke formation process. This suggestion is further

supported by experimentally observed stronger changes in the propylene selectivity than the ethylene selectivity, upon cycling on going from $Pr_2Zr_2O_7$ to $Yb_2Zr_2O_7$ (Figure 13).

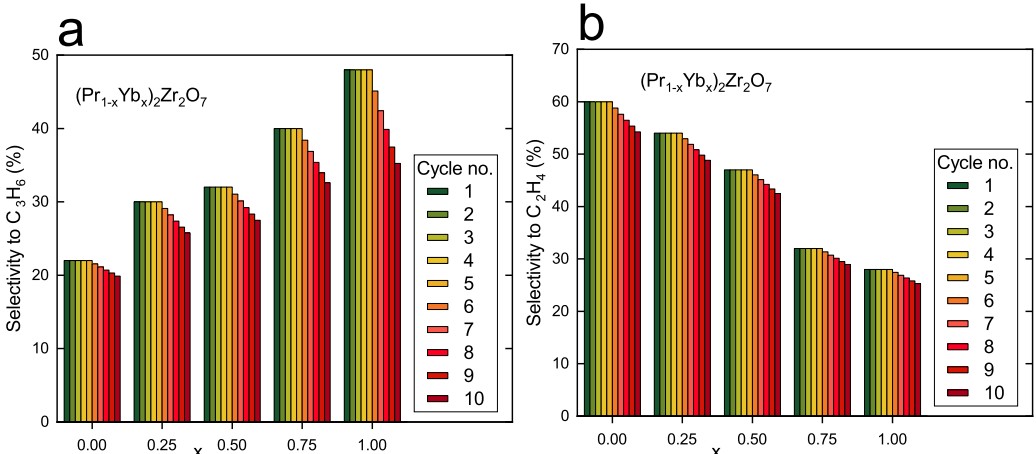

**Figure 13.** Changes in the propylene (**a**) and ethylene (**b**) selectivity upon cycling for the $(Pr_{1-x}Yb_x)_2Zr_2O_7$-1400 catalysts.

This poisoning or surface blocking is non-specific. The catalyst activity can be restored via oxidative regeneration, aimed at the removal of blocking soot deposits but avoiding the decomposition of the catalyst structure and the degradation of appropriate active sites. The choice of regeneration conditions at a temperature of 750 °C for 100 h after every 10 cycles allows for carbon combustion to work according to the mechanism of a heterogeneous process. The specificity of the process lies in the fact that the chemical stage cannot be considered in isolation from the process of transferring a gaseous oxidizer (air oxygen) from the surrounding space to the surface of a burning solid. The combustion rate depends on the chemical properties of carbon and the regeneration characteristics. Oxygen supply to the combustion zone is carried out by diffusion and, therefore, depends on many factors: the shape and size of the coking hearth, the movement of the gas medium, the diffusion coefficients of oxygen, and the reaction products in the space above the surface of the catalyst and in cracks or pores. For this reason, we performed regeneration after 10 cycles. This avoids the formation of a solid resin, which permanently blocks the catalytic center. For the same reason, we do not increase the processing temperature with the possibility of reducing its time.

At the initial moment of burnout, this will happen due to oxygen being near its surface. After its use, a layer of combustion products $–CO_2$ is formed around the incandescent surface. The combustion rate will decrease, and the process may stop if there is no oxygen supply from more distant areas of the gas space. This flow occurs due to diffusion, and the combustion rate will be determined by the magnitude of the diffusion flow. The intensity of diffusion largely depends on the intensity and nature of the movement of the gas medium near the surface. The rate of a chemical reaction is determined mainly by temperature and obeys the Arrhenius law.

At high temperature, the carbon oxidation reaction proceeds very quickly, and the overall speed of the process will be limited by the diffusion of oxygen to the surface. Thus, we do not reduce the regeneration temperature, and the value of 750 °C is optimal. Thus, the process consists of two processes that are different in nature: the process of oxygen transfer from the gas space to the coking site and the process of its chemical interaction with the surface of solid carbon. Both of these processes are interrelated, but each of them has its own patterns. The most important of these processes is the process of oxygen consumption, which is characterized by many chemical reactions.

The mechanism of the complex reaction of an oxygen–carbon compound consists in the formation of two oxides of CO and $CO_2$ simultaneously through an intermediate physico-

chemical complex of the $C_xO_y$ type, which is then split into CO and $CO_2$. Accordingly, the equation of the carbon combustion reaction can be written as follows:

$$hC + uO_2 \rightarrow mCO + nCO_2$$

Then, a homogeneous combustion reaction proceeds with the release of carbon monoxide:

$$2CO + O_2 \rightarrow 2CO_2$$

This reaction can occur both near the surface of coal and inside the coal mass, in its pores and cracks. Another reaction is a heterogeneous reaction between hot coal and carbon dioxide:

$$C + CO_2 \leftrightarrow 2CO$$

This happens at a noticeable rate in places where there is a shortage of oxygen but where the carbon temperature is high enough. We found that the treatment of the catalysts in an air flow for 100 h at 750 °C gives rise to the full recovery of catalytic activity and complete removal of undesired soot, which is evidenced by a nearly 100% carbon balance (Table 5). The amount of free carbon formed and burned out was determined by weighing the reactor. Since the "carbon balance" has good convergence, this allowed us to consider this method very accurate.

## 4. Conclusions

The local atomic and crystal structures of praseodymium/ytterbium zirconates with the common formula $(Pr_{1-x}Yb_x)_2Zr_2O_7$ ($0 \leq x \leq 1$) synthesized via the coprecipitation followed by the calcination are elucidated in detail with the use of a set of diffraction, spectroscopic, and electron microscopy techniques. All of the precipitated precursors are found to be hydrated basic carbonates $(Ln/Zr)(OH)_5(CO_3) \cdot nH_2O$ ($1.5 < n < 2.5$). They are all X-ray amorphous irrespective of the metal cation composition. The initial crystallization of the precursors at 600–800 °C results in the formation of nanocrystalline powders with the defect fluorite structure. The calcination at a higher temperature 1100–1200 °C gives rise to the complete removal of carbonates, which induces a chain of phase transitions fluorite (sp. gr. $Fm\bar{3}m$) → pyrochlore (sp. gr. $Fd\bar{3}m$) at $0 \leq x \leq 0.5$. The $(Pr_{0.25}Yb_{0.75})_2Zr_2O_7$ sample retains the defect fluorite structure up to the maximum calcination temperature 1400 °C. A complex oxide with a specific structure differing from cubic fluorite is formed in the case of $Yb_2Zr_2O_7$ calcined at 800 °C. Its calcination at even higher temperature affords the emergence of the rhombohedral $\delta$-phase (sp. gr. $R\bar{3}$).

We clearly demonstrate that the peculiarities of crystal and the local atomic structures of the $(Pr_{1-x}Yb_x)_2Zr_2O_7$ ($0 \leq x \leq 1$) samples prepared by calcination at 1400 °C/3 h exert an essential influence on their catalytic activity as regards the catalytic cracking of propane. The catalytic activity quantified via the conversion of propane is strongly affected by the total number and strength of accessible Lewis acid surface sites on $(Pr_{1-x}Yb_x)_2Zr_2O_7$ as identified by the pyridine adsorption measurements. More specifically, the progressive replacement of $Pr^{3+}$ with $Yb^{3+}$ cations leads to an increase in the number of electron acceptor centers, which results in increased propane conversion. Meanwhile, an opposite trend in the product selectivity (ethylene vs. propylene) is observed with variation of the catalysts' composition and structure. Indeed, the ethylene (formed due the propane decomposition via the C–C bond scission) selectivity is decreased and the propylene (formed due to the propane dehydrogenation) is increased on going from $Pr_2Zr_2O_7$ (sp. gr. $Fd\bar{3}m$) to $Yb_2Zr_2O_7$ (sp. gr. $R\bar{3}$). We elaborate a tentative mechanism stating that it is the geometry match between the metal–oxygen (Pr–O, Yb–O, and Zr–O) bond lengths in the active sites and the adsorbed propane molecule size that is the key factor governing the dominant route of catalytic propane cracking.

**Supplementary Materials:** The following supporting information can be downloaded at https://www.mdpi.com/article/10.3390/cryst13091405/s1, Figure S1: SEM images of PrZrOH-prec (a), $Pr_{0.5}Yb_{0.5}ZrOH$-prec (b) and YbZrOH-prec (c) particles; Figure S2: The STA curves of lanthanide zirconate precursors; Figure S3: FT-IR spectra of $Pr_2Zr_2O_7$ powders prepared at different temperatures; Figure S4: Raman spectra $Pr_2Zr_2O_7$ (a), $(Pr_{0.5}Yb_{0.5})_2Zr_2O_7$ (b), and $Yb_2Zr_2O_7$ (c) powders prepared at different temperatures; Table S1: The results of XRD Rietveld refinement for $(Pr_{1-x}Yb_x)_2Zr_2O_7$ prepared at different temperatures; and Table S2: Fractional atomic coordinates, isotropic displacement parameters, and fractions of antisite defects ($Ln_{Zr}+Zr_{Ln}$) for $(Pr_{1-x}Yb_x)_2Zr_2O_7$ samples calcined at 1400 °C.

**Author Contributions:** Conceptualization, V.V.P. and E.B.M.; methodology, V.V.P. and E.B.M.; validation, V.V.P., A.P.M., E.B.M. and Y.V.Z.; formal analysis, A.A.Y., B.R.G., O.V.C., S.G.R. and E.B.M.; investigation, V.V.P., E.B.M., Y.V.Z., S.G.R., M.M.B., A.A.P., E.S.K., N.A.K., E.V.K., V.N.K., I.V.S., N.A.T., O.N.S. and N.V.O.; resources, V.V.P.; data curation, A.A.Y., B.R.G. and O.V.C.; writing—original draft preparation, V.V.P., E.B.M. and Y.V.Z.; writing—review and editing, V.V.P., E.B.M., Y.V.Z. and A.A.I.; visualization, E.B.M., A.A.Y., B.R.G., O.V.C., S.G.R. and A.A.I.; supervision, A.P.M.; project administration, V.V.P. and A.P.M.; and funding acquisition, A.P.M. All authors have read and agreed to the published version of the manuscript.

**Funding:** The synthesis, synchrotron XRD, and XAFS measurements were partially supported by the Ministry of Science and Higher Education of the Russian Federation (Agreement No. 75-15-2021-1352). Raman and FT-IR measurements, SEM-EDS, ICP-AES, and STA were partially supported by the Ministry of Science and Higher Education of the Russian Federation (project number FSWU-2023-0070).

**Data Availability Statement:** Data sharing not applicable.

**Acknowledgments:** The authors acknowledge The European Synchrotron Radiation Facility (ESRF) for providing the opportunity of XAFS measurements and G.R. Castro (BM25-SpLine ESRF) personally for his help with the XAFS experiments.

**Conflicts of Interest:** The authors declare no conflict of interest.

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
