# Peer review of "Influence of Synthesis Conditions on the Crystal, Local Atomic, Electronic Structure, and Catalytic Properties of (Pr1−xYbx)2Zr2O7 (0 ≤ x ≤ 1) Powders"

_crystals, doi:10.3390/cryst13091405_

Round 1
Reviewer 1 Report
The idea behind the paper and execution are excellent. The manuscript deals with a very important issue, namely: purposeful synthesis of materials which are designed to differ by changes of the dopant ratio in a two-dopant zirconia system to determine the influence of this parameter on the properties of the obtained solids. It is logical and well-written. However, the authors didn’t put a lot of effort into writing the Introduction and Experimental sections. The Introduction lacks the context of the study, previously studied systems and the conclusions which are already available in the literature and hence does not explain the choice of dopants, other than they have not been studied before. The Experimental section does not provide proper characterization procedures.
Therefore, I recommend a minor revision, which includes addressing the following issues:
1. The Introduction needs to be substantially expanded. It lacks information about systems which contain Pr-doped zirconia and Yb-doped zirconia, which are available in the literature. Although this particular combination of dopants has not been published thus far and hence is interesting, there are many doubly promoted zirconate systems with one of the two chosen dopants. Without a wider context, the authors are misleading the reader. “Vivid interest” of researchers is summed up with only two references, whereas applications such as barrier coatings and ion conductors only have one reference. Please look into the literature to find that many papers have been published on these topics.
2. The Characterization section is lacking a lot of the important information. In some cases, it only states the apparatus used and some temperature programs. It does not provide sufficient information to perform any of the tests. This is particularly important for catalytic activity tests, which are meaningless without providing the specifics of the reaction. Please address each of the following:
ICP-AES: how were the solids treated prior to tests, i.e. what was used for digestion? What were the conditions of digestion? Was an internal standard used? If so, which one? What were the details of the measurement: flow rates, lines used for quantification, etc.
STA analysis: flow rate?
SEM: working distance? Spot size/current?
Porosity: time at 400°C?
UV absorption: calibration curve? External/internal standard? How were the samples prepared for the measurement?
Catalytic tests: flow rate, inlet gas composition, purity of gases used, type of reactor used/material from which reactor was made, volume of catalytic bed/mass of catalyst used
3. In the Results section, the pdf numbers of the XDR standards are not mentioned. Please provide these.
Minor issue:
Please place the Figures closer to their descriptions (i.e., after their first mention in the text); this will make the results easier to navigate.
Minor linguistic issues:
Line 195 “purochlore” should be “pyrochlore”
Please combine the two sentences in lines 152-153 with the next paragraph. Two sentences are not a paragraph in English and therefore should be a part of a proper paragraph.
Line 441: “featurign” should be “featuring”
Author Response
- We agree with the Reviewer and thus we have made appropriate changes and additions to the Introduction and extended the list of References accordingly.
-
We agree with the Reviewer and thus we have made appropriate additions to the Characterization section addressing all mentioned points.
It should be noted that we did not use the UV absorption spectroscopy in our work. - We agree with the Reviewer and thus we have added numbers of JCPDS cards for the crystalline reference compounds mentioned in the Results section.
- {Minor issues} We agree with the Reviewer and thus we have made all necessary corrections to the text. The typos have been corrected.
Reviewer 2 Report
This paper is an excellent piece of work that reveals the close correlation between catalytic activity and crystal structure in (Pr1−xYbx)2Zr2O7 powders, particularly the geometric matching of metal-oxygen bonds within active sites. It has been found that chemical composition and calcination temperature are two main factors determining the phase composition, crystallography, and local structural parameters of these polycrystalline materials. This paper proposes a mechanism for catalytic propane cracking, which takes into account the geometric matching between the lengths of metal-oxygen bonds within the active sites and the size of adsorbed propane molecules. This is an interesting finding that further enhances our understanding of this reaction.I think this paper should ultimately be accepted, but it needs some revision before it is ready for acceptance.
The following major points must be addressed before acceptance:
1) In Part 1. Introduction, the author mentioned that (Pr1−xYbx)2Zr2O7 has recently been successfully tested for catalytic cracking of propane. However, this paper did not provide further description of this catalyst, which would enable readers to have a clearer understanding of its properties, characteristics, and advantages compared to other catalysts. It would be beneficial for the author to provide additional information about the catalyst in order to address this gap in knowledge.
2) The specific surface area data for multiple catalysts are provided by the authors in Table 4, which serves as the basis for constructing Figure 8(a). However, the calculated surface area and average pore size obtained under such low conditions have no practical significance, and the resulting function based on these is also of no value. The authors need to make corresponding modifications based on this issue.
3) The authors mentioned the crystal phase transition and the resulting increase in oxygen vacancies because of the replacement of Pr3+ with Yb3+ cations, which promotes the horizontal orientation of adsorbed propane molecules. However, the paper lacks calculations of oxygen vacancy concentration and data correlating it with reaction data. This aspect should be addressed and supplemented in the study.
4) In Part 3.2. Catalytic Properties, The author mentioned that treatment in air at 750°C for 100 hours can lead to full recovery of catalytic activity and complete removal of undesired soot. However, there is a lack of corresponding reaction data to support this claim.
5) It would be beneficial for the author to include quantitative data related to catalyst preparation and reaction in both the abstract and the conclusion of the paper. Including such data would enhance the clarity of the study, allowing readers to assess the experimental results and conclusions more effectively.
Author Response
- We agree with the Reviewer and thus we have extended the respective section of the manuscript. In particular, the respective sentence now reads: the polycrystalline (Pr1−xYbx)2Zr2O7 used as catalysts in [Catalysts 2023, 13, 396] were synthesized at 1000°C. Based on our previous results [J. Phys.: Conf. Ser. 2017, 941, 012079; J. Alloys Compd. 2020,832, 154863] all studied samples had the same disordered defect-fluorite structure. We cannot provide more detailed additional information, since we currently do not have these samples in stock.
-
We agree with the Reviewer that the calculated surface area and average pore size are extremely small, as stated in lines 372-373 “the available surface exerts the minimum influence on the apparent activity of the catalysts towards the propane dehydrogenation”. This is a very important note because it allows you to:
1) Compare these catalytic systems with similar catalysts of the same composition, but obtained at a lower temperature (1000°C) and having higher specific surface areas (21-44 m2/g) [Catalysts 2023, 13, 396], and thus unambiguously establish the nature of catalytic centers.
2) This fact allows us to assert that the total and specific activity pass through the maxima with a very small degree of filling. Each real surface of these catalytic systems is characterized by a blocky, mosaic structure, as a result of which isolated migration regions may appear on the surface, separated from each other by energy or geometric barriers. They can be similar to real cells, for example, faces of elementary crystals adjacent to adsorption centers with an increased adsorption potential characteristic of an energetically inhomogeneous surface. Cracks and other surface disturbances, crystal defects, and stoichiometric composition disorders can also cause migration areas. This leads to the appearance of random catalytic centers. If there were a large number of them, we would not be able to unambiguously characterize the mechanism of the processes taking place. For the formation of active centers in the form of a cluster of n-atoms, it does not matter what the origin and nature of these inhomogeneities are. It is important that when forming a layer of catalytic centers, the surface (which is very insignificant) allows free migration of particles only in limited areas. These surface disturbances are an obstacle to free movement through the volume of the catalyst, forming potential pits where additional catalytic centers should accumulate. In our case, their number is very small. The catalyst surface is a collection of closed migration regions. Experimental data on the sintering of catalysts were a confirmation of these ideas. The rate of deactivation of catalysts obeys the first-order equation for the concentration of additional centers on the surface. Consequently, the additional centers on the surface do not depend on each other and do not interact with each other. The size of the migration areas significantly exceeds the radius of action of molecular forces (by tens of times). Consequently, the formation of additional centers of catalytic cents is an independent event. The distribution of these cents over the surface of the catalyst obeys the law of chance and tends to a minimum. Thus, we can make an unambiguous conclusion that the main catalytic centers are precisely the Lewis acid centers.
Relevant comments are included in the text of the manuscript. -
Regarding this Reviewer’s comment, we consider it necessary to respond as follows:
In the work discussed, the determination of the parameters of the structural basis was carried out using the Rietveld refinement based on powder X-ray diffraction data. This technique involves clarifying the required structural parameters taking into account the specific atomic scattering factor for each atomic position. The X-ray scattering factor for oxygen atoms is 5-8 times less than the same parameter for heavy Zr/Pr/Yb atoms. This affects the very weak sensitivity of the chosen method when clarifying the parameters of filling oxygen positions and the impossibility of reliably determining these parameters together with the parameters of filling cation positions.
However, it should be noted that the total concentration of oxygen vacancies in (Pr1−xYbx)2Zr2O7 weakly depends on the type of cubic pyrochlore/fluorite structure. The ratio of oxygen vacancies to occupied oxygen positions is ensured by the stoichiometry and electrical neutrality of the compound. In both structures, this parameter is at the level of one vacancy per seven filled oxygen positions. When the type of crystal structure changes, only the spatial distribution of oxygen vacancies changes: from the precisely specified position of oxygen vacancy 8a (1/8, 1/8, 1/8) in pyrochlore (56 filled oxygen positions out of 64 possible in the unit cell) to equally probable filling with coefficient 7/8 oxygen position 8c (1/4, 1/4, 1/4) in fluorite.
Relevant comments are included in the text of the manuscript. -
Regarding this Reviewer’s comment, we consider it necessary to respond as follows:
The choice of regeneration conditions at a temperature of 750°C for 100 hours after every 10 cycles allows carbon to work according to the mechanism of a heterogeneous process. The specificity of the process lies in the fact that the chemical stage cannot be considered in isolation from the process of transferring a gaseous oxidizer (air oxygen) from the surrounding space to the surface of a burning solid. The combustion rate depends on the chemical properties of carbon and the regeneration characteristics. Oxygen supply to the combustion zone is carried out by diffusion and, therefore, depends on many factors: the shape and size of the coking hearth, the movement of the gas medium, the diffusion coefficients of oxygen and reaction products in the space above the surface of the catalyst and in cracks or pores. For this reason, we performed regeneration after 10 cycles. This avoids the formation of a solid resin, which permanently blocks the catalytic center. For the same reason, we do not increase the processing temperature with the possibility of reducing its time.
At the initial moment of burnout, this will happen due to oxygen being near its surface. After its use, a layer of combustion products – CO2 - is formed around the incandescent surface. Combustion rate will decrease, and the process may stop if there is no oxygen supply from more distant areas of the gas space.
This flow occurs due to diffusion, and the combustion rate will be determined by the magnitude of the diffusion flow. The intensity of diffusion largely depends on the intensity and nature of the movement of the gas medium near the surface. The rate of a chemical reaction is determined mainly by temperature and obeys the Arrhenius law.
At high temperature, the carbon oxidation reaction proceeds very quickly, and the overall speed of the process will be limited by the diffusion of oxygen to the surface.
Thus, we do not reduce the regeneration temperature, and the value of 750°C is optimal.
Thus, the process consists of two processes that are different in nature: the process of oxygen transfer from the gas space to the coking site and the process of its chemical interaction with the surface of solid carbon. Both of these processes are interrelated, but each of them has its own patterns. The most important of these processes is the process of oxygen consumption, which is characterized by many chemical reactions.
The mechanism of the complex reaction of an oxygen-carbon compound consists in the formation of two oxides of CO and CO2 simultaneously through an intermediate physico-chemical complex of the CxOy type, which is then split into CO and CO2.
Accordingly, the equation of the carbon combustion reaction can be written as follows:
hC + uO2 → mCO + nCO2
Then a homogeneous combustion reaction proceeds with the release of carbon monoxide:
2CO + O2 → 2CO2
This reaction can occur both near the surface of coal and inside the coal mass, in its pores and cracks.
Another reaction is a heterogeneous reaction between hot coal and carbon dioxide:
C + CO2 ↔ 2CO
This happens at a noticeable rate in places where there is a shortage of oxygen, but where the carbon temperature is high enough.
“Carbon balance” (%), which allows us to estimate the proportion of formed amorphous carbon participating in the process of surface decontamination, was calculated using the formula:
The amount of free carbon formed and burned out was determined by weighing the reactor. Since the “carbon balance” has good convergence, this allowed us to consider this method very accurate.
Relevant comments are included in the text of the manuscript. - We agree with the Reviewer and thus we have made all necessary additions to both the Abstract and the Conclusion sections.